# Minimum 10-Year Results of Modular Metal-On-Metal Total Hip Arthroplasty

**DOI:** 10.3390/jcm11216505

**Published:** 2022-11-02

**Authors:** Hiroki Wakabayashi, Masahiro Hasegawa, Yohei Naito, Shine Tone, Akihiro Sudo

**Affiliations:** Department of Orthopaedic Surgery, Mie University Graduate School of Medicine, 2-174 Edobashi, Tsu 514-8507, Mie, Japan

**Keywords:** long-term results, metal-on-metal, pseudotumors, total hip arthroplasty

## Abstract

Background: this study aimed to assess the long-term outcomes of (a minimum of 10-years) total hip arthroplasty with a metal-on-metal acetabular prosthesis. Methods: Eighty-nine primary total hip arthroplasties (82 patients) were performed using a Pinnacle modular metal-on-metal acetabular prosthesis. Clinical hip function outcomes were evaluated using the Japanese Orthopaedic Association hip score preoperatively and at the final follow-up. Radiological analysis was performed at the final follow-up and magnetic resonance imaging in all hips postoperatively. Results: Out of 82 patients, 17 were excluded who were followed up for <10 years. Of the remaining 65 patients (70 hips), 19 (20 hips) developed pseudotumors during 2–10 years postoperatively. After 10 and 13 years, the survival rates of revision endpoint were 93.6% and 90.4%, respectively. Clinical hip function outcomes had improved significantly at the final follow-up. In the radiological analysis, the mean cup angle of inclination and mean ratio of femoral offset on the operated hip to the contralateral hip was highest in patients with revision surgery for adverse reactions to metal debris. Conclusions: This study showed a 29.0% prevalence of pseudotumors. Some cases required revisions even after 10 years following surgery. Regular clinical surveillance is recommended for the early detection of adverse reactions to metal debris.

## 1. Introduction

Total hip arthroplasty (THA) is performed in patients with end-stage osteoarthritis of the hip mainly to relieve long-term pain [1,2]. However, despite improved implant designs and surgical techniques, aseptic loosening and osteolysis due to particulate debris generated by conventional polyethylene have become major limitations to prosthetic long-term survivorship.

In 2000, there was an increasing trend of implantation of large diameter and hard bearing surface prostheses in THA, particularly metal-on-metal (MOM), with an estimated 1 million hips implanted worldwide [3].

MOM prostheses have perceived benefits such as improved arc of motion, decreased risk of dislocation, lower volumetric wear, and durability of bearing surfaces. However, despite the potential advantages, a continuing concern with MOM articulation is the release of metal ion debris locally and systemically in the patients’ blood and urine. Adverse reactions to metal debris (ARMD) include the formation of pseudotumors (PTs), metallosis, and soft tissue necrosis. High short-term failure rates have been reported for various MOM THA owing to ARMD [4]. A previous study reported that the use of cobalt chromium molybdenum (CoCrMo)-on-(CoCrMo) could potentially increase contact pressure by more than 47% compared to that of titanium alloy (Ti6Al4V) metal-on-metal couple bearings in the model based on finite element simulation [5]. Ammarullah MI et al. reported that the Tresca stress value rose, and the stress distribution widened by increasing the body mass index under normal walking condition using a 2D finite element representing a CoCrMo-on-CoCrMo hip implant [6]. However, studies on the long-term outcomes of patients with MOM THA, particularly concerning ARMD and revision rates, are limited. Even if MOM bearing surfaces are no longer used, long-term data could help in defining the course and best management for these patients [7].

This study aims to investigate the long-term clinical outcomes, up to a minimum of 10 years, of total hip arthroplasty with a metal-on-metal acetabular prosthesis and examine radiological findings including magnetic resonance imaging (MRI) in Japanese patients with a 36-mm MOM THA.

## 2. Materials and Methods

In this retrospective study, we followed up patients with cementless MOM THA, who received implants between 2006 and 2010 at a single institution. Here, only the Pinnacle THA system was included.

The modular acetabular component used was a Pinnacle-A (DePuy, Warsaw, IN, USA), which consisted of a metal liner (Ultamet; DePuy, Warsaw, IN, USA). The shell was a Pinnacle hydroxyapatite-coated hemi-spherical implant with a self-locking peripheral taper to accept an Ultamet cobalt-chrome 36-mm (cup size: 52 mm and over) or 28-mm (cup size: 50 mm) inner diameter metal liner. All patients received a 36-mm or 28-mm Articul/eze femoral cobalt-chrome metal head (DePuy, Warsaw, IN, USA). In all the MOM THAs, an S-ROM stem (DePuy, Warsaw, IN, USA) was used for the femoral stem. All surgical procedures were performed using the same equipment and a posterolateral approach.

Regarding primary THA, 89 primary THAs (82 patients: 75 women and 14 men) were performed using a Pinnacle (DePuy, Japan) modular MOM acetabular prosthesis. The preoperative diagnosis of most patients was osteoarthritis. The patients’ mean age at surgery was 62.6 years, the mean body weight was 56.7 kg, and the mean body mass index (BMI) was 24.2 kg/m^2^ (Figure 1). The Japanese Orthopaedic Association (JOA) hip score was used to assess the clinical outcomes of the patients who were followed up for a minimum of 10 years. Anteroposterior (AP) and lateral hip radiographs were taken for each patient and analyzed by an experienced radiologist and orthopedic surgeon. The exclusion criterion was a follow-up period of <10 years.

This study was approved by our Institutional Review Board, and all the patients who participated in the study provided informed consent. All procedures were in accordance with the ethical standards of the institutional committee and with the 1964 Helsinki Declaration and its later amendments.

### 2.1. Clinical Evaluation

Clinical hip function outcomes were evaluated using the JOA hip score preoperatively and at the final follow-up. The JOA hip score consists of the following four subcategories: pain (up to 40 points), range of motion (ROM, 20 points), ability to walk (walk, 20 points), and activities of daily living (ADL, 20 points). A perfect JOA hip score is 100; the worst is 0 [8].

### 2.2. Radiological Evaluation

AP and lateral radiography of the operated hip were performed at the follow-up visit to evaluate implant loosening. Osteolysis was determined using the Gruen and DeLee classifications [9,10]. Additionally, the inclination angle of the acetabular component was measured [11]. Femoral offset (FO) was measured as the distance from the center of rotation of the femoral head to a line dissecting the long axis of the femur [12]. Additionally, the ratio of the FO (RFO) on the operated hip to the contralateral hip was calculated. The MRI for PT screening following the MOM THA was first performed 2 years postoperatively, and thereafter every 2 to 3 years until 10 years postoperatively.

### 2.3. Statistical Analysis

Statistical analyses were performed using the nonparametric Wilcoxon signed-rank test, analysis of variance, and Spearman’s rank correlation coefficient. A *p* value of <0.05 was considered significant. Kaplan–Meier survivorship analysis was performed using revision for any reason as the end point including patients who died before the 10-year follow-up and the survivorship of revision and loss of follow-up (including transfer to another hospital and not visited). All statistical analyses were performed using IBM SPSS Statistics (IBM Japan, Tokyo, Japan).

## 3. Results

Seventeen patients (19 hips) were lost before their 10-year follow-up. One patient (both hips) died owing to pneumonia after 2.8 years. Two patients (2 hips) died owing to heart failure after 3.6 and 4 years. Three patients (3 hips) died owing to medical disease, cerebral aneurysm, or breast cancer after 7, 9, and 9.5 years, respectively. Two patients (3 hips) transferred to another hospital after 9 and 9.5 years because visiting our hospital was difficult. The remaining nine patients (9 hips) did not visit before their 10-year follow-up for unclear reasons. After excluding patients followed up for <10 years, 65 patients (70 hips) were evaluated at the mean 13-year follow-up (range, 10–15 years). Patient demographics are depicted in Figure 1. The indications for primary THA were primary hip osteoarthritis (OA) in 89% (59/70 hips) and rheumatoid arthritis in 12.9% (9/70 hips) patients. One patient (1/70 hips) was diagnosed with avascular necrosis of the femoral head and secondary hip OA in post-traumatic fracture. We included 65 hips that received a 36-mm-diameter Articul/eze femoral head and 5 that received a 28-mm-diameter head.

Twenty hips (29%) were observed for PTs attributable to the MOM articulation by MRI from 2 to 10 years postoperatively with a mean of 5.88 (±standard deviation (SD), 2.64) years. Seven hips (7 patients) were switched to a metal-on-polyethylene articulation from 5.1 to 15 years postoperatively owing to pain, swelling, and/or implant failure. A 36-mm head was used in all seven hips, and six of the patients were females. PTs were identified in six hips; one hip had none (Case 4), although all hips were diagnosed with ARMD.

Another patient, who underwent a revised hip arthroplasty due to ARMD with PT, had cup loosening 8.1 years after primary arthroplasty (Case 1). Dislocation was noted in 6/70 hips (6 patients). Two hips that had rebound dislocation with PT were revised due to ARMD 5.8 and 15 years after primary arthroplasty (Cases 2 and 7). One hip revised due to ARMD with PT was observed to have cup osteolysis (zones 1 and 3) and trochanteric region osteolysis (zones 1 and 7) (Case 6). Another hip that had revised metal-on-polyethylene articulation for PT was infected with Listeria monocytogenes (Case 5). The stem and liner of the infected THA with PT were removed, and irrigation and debridement (I&D) with modular component exchange were performed (Table 1). Some cases (Case 5, 6, and 7) required revisions even after 10 years following THA.

One patient (1 hip) was infected with *Staphylococcus aureus* 3.5 years after primary arthroplasty. The infected THA eventually led to removal of the stem and liner, and revision THA with metal-on-polyethylene articulation by I&D with modular component exchange was performed. The overall implant survival rates at 10 and 13 years were 93.6% and 90.4%, respectively (Figure 2a). The rates of survivorship of revision and loss of follow-up at 10 and 13 years were 82.2% and 79.3%, respectively (Figure 2b).

We assessed radiological outcomes in 64 patients (69 hips), excluding one hip that was infected 3.5 years after primary arthroplasty. The mean acetabular cup inclination was 46.8° (±SD, 6.9°), the mean anteversion was 19.3° (±6.0°), and the mean RFO on the operated hip to the contralateral hip was 0.914 (±0.256). The stress shielding progressed to grade 3 or 4 in 27 hips (39.1%) during the study period.

Forty-eight hips had no PTs and 14 had PTs but no revision surgery. Seven patients (7 hips) underwent revision surgery for ARMD. In the radiological analysis, the mean cup angle of inclination was the highest in the patients with revision surgery for ARMD (57.8°) compared with the patients with PTs but no revision surgery (47.7°) and those with no PTs (46.3°). The mean RFO was significantly higher in the patients with revision surgery for ARMD (1.043) than that of the patients with PTs but no revision surgery (0.787) (Table 2). The mean RFO in the patients with no PTs (0.932) was not significant compared with the patients with revision surgery for ARMD and those with PTs but no revision surgery.

The mean (±SD) preoperative JOA hip score of 46.0 (±11.6) improved significantly to 82.2 (±12.7) postoperatively at the final follow-up (*n* = 62 hips, excluding 8 revision cases) (Figure 3). No significant differences in JOA hip scores (mean ± SD) were observed between patients without PTs (81.0 ± 14.0), those with PTs but no revision (82.4 ± 13.4), and those treated with revision THA for ARMD (82.0 ± 10.9).

## 4. Discussion

THA with a large head has perceived benefits, including improved arc of motion and decreased risk of dislocation [13]. However, a large diameter for MOM THA has generally not lived up to clinical expectations owing to unacceptably high revision rates of large head MOM conventional THA within 10 years of implantation. They have their own inherent limitations, such as adverse local tissue reaction (ALTR), ARMD, and PTs [2].

Failure of MOM THA and ALTR were recognized only after these replacements had been in use for several years [3,14]. The 36-mm MOM Pinnacle THA system (DePuy, Leeds, United Kingdom) has been commonly implanted worldwide. Moreover, several large cohort studies examining the 36-mm MOM Pinnacle THA system have confirmed these moderate-to-high survival rates. In a large cohort studies, Langton et al. [15] over 9 years, Kindsfater et al. [16] over 9 years, and Pearce et al. [17] over 10 years have identified survival rates of 84%, 94.4%, and 83.4%, respectively. Furthermore, LaHaise et al. have suggested that the survivorship of MOM hips may be relatively high (5.4 revisions per 1000 person-years) [18]. Several small but long-term single-center cohort studies examining the 36-mm MOM Pinnacle THA system have confirmed an 83% survival rate after 15 years [7]. In this study, the overall implant survival rate at a mean of 13 years was 90.4%; 10% of the revisions were because of ARMD.

The Pinnacle-A shell is a hydroxyapatite-coated hemi-spherical implant approved in Asia. In a previous report of mid-term results, MOM THA with Pinnacle-A had a higher incidence of osteolysis than that of ceramic-on-ceramic THA. However, no significant difference was observed in the 8-year survival rates between implants when using implant loosening and revision THA as endpoints [19]. According to Higuchi et al. [19], the incidence of asymptomatic ALTR/ARMD and PTs was between 40% and 60%, and the incidence of symptomatic (pain or discomfort) PTs was 8.9%. In our study, 10% (7 hips) of MOM THAs required revision surgery for symptomatic ARMD, and 20.3% (14 hips) had asymptomatic PTs. This incidence rate of symptomatic PTs was practically the same, and that of the asymptomatic PTs was lower than that in previous reports. This may be because of the small cup inclination angle and offset in the asymptomatic PT group compared with the revision surgery for the symptomatic ARMD group. PTs may be derived from cell toxicity resulting from particulate wear debris, which may lead to revision THA. The previous studies have reported the risk factors of ARMD.

Regarding implant-related factors, PTs have been reported to mainly occur in patients with MOM with large-diameter heads [20]. Increased head size has been reported to increase the horizontal lever arm, leading to increased taper wear rates in MOM bearings ≥36-mm [21], with increasing retrieval studies observing higher corrosion scores with the use of larger femoral heads [22]. Moreover, high femoral offset stems were significant predictors for all-cause revision in the 36-mm Corail Pinnacle MOM THA implant [17].

Regarding surgical factors, acetabular cup inclination >50° has been positively correlated with increased serum metal ion levels [23]. The reason may be because of edge loading and high wear [24]. Potential patient factors associated with increased failure rates in MOM hip arthroplasty include female sex in a systematic review [25] and dysplasia in retrospective study [26]. Furthermore, other possible patient factors reported were metal sensitivity [27], low BMI [28], and low activity levels [29].

In this study, the revision surgery group received a 36-mm head, was comprised of predominantly females, and had the highest cup inclination angle and offset. The main limitations of this study include its retrospective design, limited sample size, absence of a control group, high dropout rate, and two-dimensional (2D) radiological evaluation. Our previous study reports the usefulness and accuracy of the three-dimensional (3D) method, in line with previous reports [30,31]. In this study, no CT data was included. In the previous study, the reported comparison between the accuracy of anteroposterior radiographs (2D) and that of three-dimensional computed tomography (3D-CT) scans found a mean difference of 0.6° for inclination and a mean difference of 0.1° for anteversion between 3D CT scans and plain radiographic measurements. The means for absolute differences were 3.1° for inclination and 7.2° for anteversion [31]. We believe that the assessment of cup position is acceptable with a 2D method similar to that of previous reports. Patient-specific factors such as contractures or leg rotation that depend on patient compliance impede the comparability of radiographic images. We compared the ratio of the femoral offset on the operated hip to the contralateral hip for minimizing such projection errors.

The highlights of our study include the high cup inclination angle and offset and the fact that all patients in the revision surgery group received a 36 mm head. This study is novel as the long-term outcomes of Japanese patients with MOM THA over 10 years after the procedure is reported. Compared with previous studies, this study reports the requirement of revision surgery for ARMD even after 10 years following THA. In future, follow-up for a longer period than that of our study and prompt treatment of any revision cases will be necessary. Patients at high risk of revision need to be prevented from dropping out.

## 5. Conclusions

Clinical scores such as the JOA score revealed good outcomes at the mean 13-year follow-up. However, the prevalence of PTs per hip was 29.0%. Some cases required revisions even after 10 years following THA. Although this MOM THA has not failed as dramatically as other similar designs, we recommend against its continued use and advise regular clinical surveillance, such as MRI, for early detection of ARMD.

## Figures and Tables

**Figure 1 jcm-11-06505-f001:**
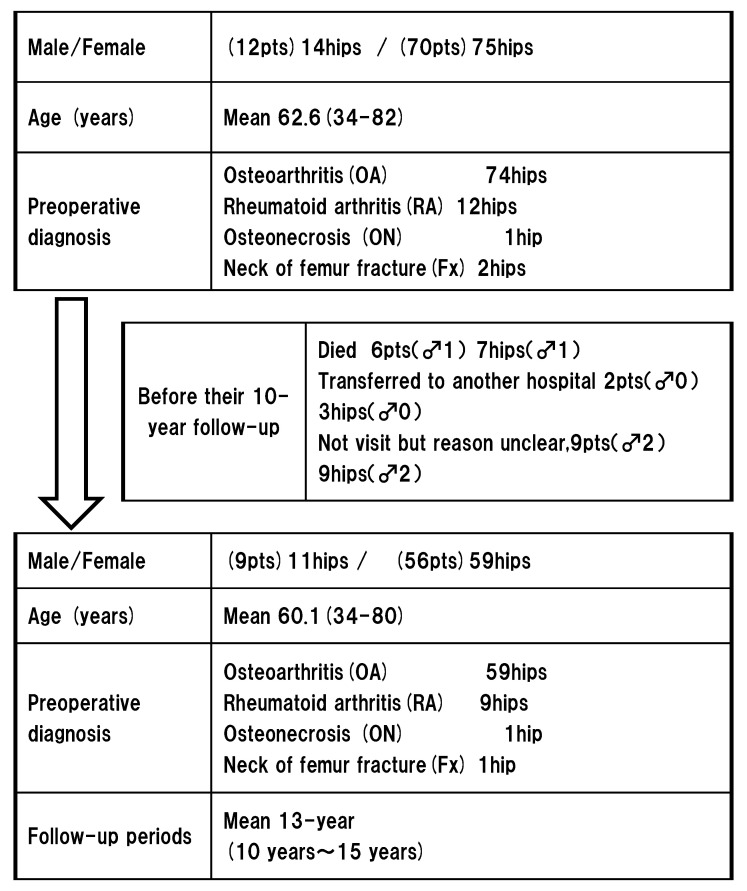
Study flow chart of patient selection and demographic background.

**Figure 2 jcm-11-06505-f002:**
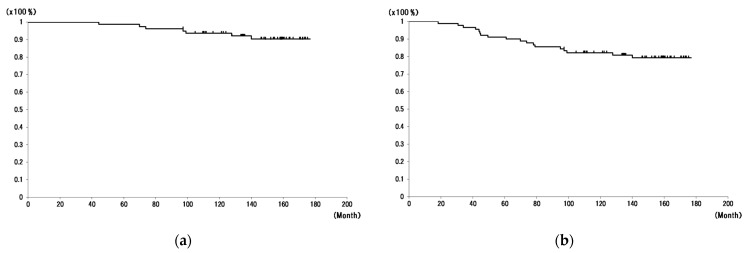
Implant and follow-up survival. The survivorship of revision endpoint (**a**) and the survivorship of revision and loss of follow-up (including transfer to another hospital and no visit) (**b**).

**Figure 3 jcm-11-06505-f003:**
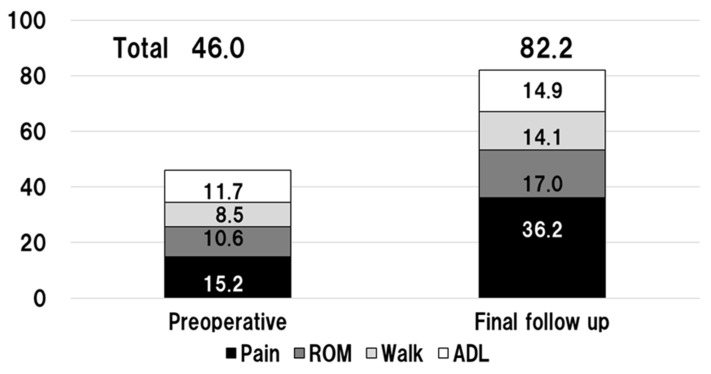
Clinical hip function outcomes using the Japanese Orthopaedic Association (JOA) hip score preoperatively and at the final follow-up.

**Table 1 jcm-11-06505-t001:** Revision cases. Seven hips were switched to a metal-on-polyethylene articulation from 5.1 years to 15 years postoperatively because of pain and swelling. Fx, secondary hip OA in post-traumatic fracture; OA, osteoarthritis; RA, rheumatoid arthritis.

Case	Age and Sex	Diagnosis	BMI	Observed PT after Primary Arthroplasty	Symptom	Time to Revision (Years)	Cup Inclination	Cup Anteversion
1	63 female	Fx	23.6	5-year	pain, cup loosening	8.1-year	64.4	13.8
2	57 female	OA	23.6	3-year	pain, dislocation	5.8-year	62.0	19.0
3	66 female	OA	22.8	5.1-year	pain	5.1-year	47.8	18.8
4	34 female	RA	25.2	not observed	pain	8-year	63.0	29.6
5	53 female	RA	16.5	4-year	pain, swelling	10.5-year	55.7	24.2
6	63 female	RA	21.6	5-year	swelling, osteolysis	11.5-year	55	14.0
7	67 male	OA	24.1	6-year	pain, dislocation	15-year	56.6	17.9

**Table 2 jcm-11-06505-t002:** Evaluation among hips without pseudotumors, with pseudotumors but no revision surgery, and with revision surgery for adverse reactions to metal debris. ARMD, adverse reactions to metal debris; PT, pseudotumor; PT−, hips without PTs; Not rev. with PT+, hips with PTs but no revision surgery; Revision for ARMD, hips with revision surgery for ARMD.

	PT- (48 Hips)	Not Rev. with PT+ (14 Hips)	Rev. for ARMD (7 Hips)	*p*
female: hip (%)	41(85.4%)	13(92.9%)	6 (85.7%)	0.8647
age (years)	60.4 ± 9.1	61.9 ± 7.8	57.6 ± 11.5	0.8501
BMI (kg/m^2^)	24.5 ± 4.0	24.5 ± 4.5	22.5 ± 2.8	0.3739
head diameter 36 mm: hip (%)	44 (91.7%)	14 (100%)	7 (100%)	0.7911
cup inclination angle (°)	46.3 ± 6.8	47.7 ± 7.4	57.8 ± 5.8	0.0031
cup anteversion angle (°)	19.1 ± 5.7	18.0 ± 7.6	19.6 ± 5.6	0.6833
ratio of femoral offset	0.932 ± 0.071	0.787 ± 0.048	1.043 ± 0.024	0.0463
observed PT after primary arthroplasty (years)		6.4 ± 2.8	4.2 ± 1.3	0.3025
JOA score at final follow up	81.0 ± 14.0	82.4 ± 13.4	82.0 ± 10.9	0.9195

## Data Availability

Not applicable.

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
