# Peer review of "Minimum 10-Year Results of Modular Metal-On-Metal Total Hip Arthroplasty"

_jcm, 2022, doi:10.3390/jcm11216505_

Round 1
Reviewer 1 Report
Thank you for the opportunity to read this interesting paper. The authors reported the clinical results of MoM THA minimum 10-year follow up. I think this is an important paper. However, I have some minor concerns to be revised.
<Minor comments>
・Kaplan-Meier survivorship analysis
In general, Kaplan-Meier survivorship analysis should be performed including dropout patients. In this study, relatively many patients were lost before 10 years. Did the authors analyze the survivorship including these dropout patients? If not, the authors should re-analyze the survivorship.
・Fig. 2
I think there are seven crank points in Fig. 2. Eight patients were revised totally (seven ARMDs, one infection). The X axis of Fig. 2 include only 160 months. Case 7 revised 15 years (180 months) after primary THA. The authors should re-make Fig. 2 including 180 months of X axis. Further, number at risk should be included in Fig. 2. (Lost patients should be included)
・Table 2
Some p values in Table 2 were shown as ns. That’s not appropriate. The authors should describe p values as number.
Author Response
Reviewer 1
On the comment of [I think this is an important paper.]
Replies to Reviewer #1
Authors are grateful to Reviewer #1 for encouraging comments. We have revised the indicated parts of the manuscript according to the comments. Corrections in the newly revised manuscript are underlined.
Please note that your review comments are shown in italic below and our replies in non-italic.
On the comment of [In general, Kaplan-Meier survivorship analysis should be performed including dropout patients. In this study, relatively many patients were lost before 10 years. Did the authors analyze the survivorship including these dropout patients? If not, the authors should re-analyze the survivorship.]
Reply: Thank you for suitable suggestion. We re-analyzed the survivorship of revision endpoint including patients who died before the 10-year follow-up and the survivorship of revision and loss of follow-up (including transfer to another hospital and not visited).
On the comment of [I think there are seven crank points in Fig. 2. Eight patients were revised totally (seven ARMDs, one infection). The X axis of Fig. 2 include only 160 months. Case 7 revised 15 years (180 months) after primary THA. The authors should re-make Fig. 2 including 180 months of X axis. Further, number at risk should be included in Fig. 2. (Lost patients should be included)]
Reply: Thank you for suitable suggestion. We changed Figure 2 to Figure 2a as the survivorship of revision endpoint and Figure 2b as the survivorship of revision and loss of follow-up (including transfer to another hospital and not visited).
On the comment of [Some p values in Table 2 were shown as ns. That’s not appropriate. The authors should describe p values as number.]
Reply: Thank you for raising important issues. We described p values as number in Table 2.

Reviewer 2 Report
The authors report the 10-year clinical and radiological results a hip replacement with the Pinnacle modular MOM bearing type and the SROM stem.
This constitutes another study reporting on the above-mentioned hip implant type concluding that the revision rates are unacceptably high, the authors should be clear on what the study adds to the existing literature.
As is the study lacks of novelty. The radiological assessment relies on X-rays performed at last follow-up and the cup angles were measured for each patient. The pitfalls of measuring cup orientation on X-Rays are significant and widely known, the authors should acknowledge this and exclude the measurement of cup version.
Author Response
Reviewer 2
Replies to Reviewer #2
We have revised the indicated parts of the manuscript according to the comments. Corrections in the newly revised manuscript are underlined.
Please note that your review comments are shown in italic below and our replies in non-italic.
On the comment of [This constitutes another study reporting on the above-mentioned hip implant type concluding that the revision rates are unacceptably high, the authors should be clear on what the study adds to the existing literature.]
Reply: Thank you for suitable suggestion.
The clinical factors including the increased cup inclination, increased femoral offset were already well- recognized to become a risk to undergo revision surgery due to ARMD. Furthermore, metal artifact reduction sequence (MARS) MRI were already recommended to detect pseudotumor due to ARMD in patients who were treated with MoM THA. And we discussed our data with previous reports. This study is novel as long-term outcomes of over 10 years after MOM THA in Japanese patients. We have added the sentences on line 242-244.
On the comment of [As is the study lacks of novelty. The radiological assessment relies on X-rays performed at last follow-up and the cup angles were measured for each patient. The pitfalls of measuring cup orientation on X-Rays are significant and widely known, the authors should acknowledge this and exclude the measurement of cup version.]
Reply: Thank you for raising important issues.
Successful positioning of the acetabular cup is a major goal in THA because of the influence of the cup position on the postoperative range of motion, impingement, wear and, finally, patient satisfaction. It is important for suitable measuring acetabular cup position after THA. And we and previous study reported the usefulness and accuracy of the three-dimensional (3D) method [ref 31.32]. In this study, no CT data was included. In the previous study that reported the comparison between the accuracy of anteroposterior radiographs (2D) and that of three-dimensional computed tomography (3D-CT) scans found a mean difference of 0.6° for inclination and a mean difference of 0.1° for anteversion between 3D CT and plain radiographic measurements. The means for absolute differences were 3.1° for inclination and 7.2° for anteversion [ref.32]. We believe that the assessment of cup position is acceptable with a 2D method similar to that of previous reports. Patient-specific factors such as contractures or leg rotation that depend on patient compliance impede the comparability of radiographic images. We calculated the ratio of the femoral offset on the operated hip to the contralateral hip for minimizing such projection errors.
We have added the sentences on line 229-240 and references [31.32].

Reviewer 3 Report
1. As your abstract's final sentence, include a "take-home" message.
2. Sort the keywords according to alphabetical order.
3. It is unclear whether the author's something new in this work. According to evaluation, several published studies by other researchers in the past adequately explain the issues you made in the present paper. Please be careful to highlight in the introduction section anything really innovative in this work.
4. Previous study related needs to explain in the introduction section consisting of their work, their novelty, and their limitations to show the research gaps that intend to be filled in the present study.
5. Line 33, minimal volumetric wear. Compared to what bearing couple? It is impossible if compared with ceramic-on-ceramic.
6. The authors need supporting reference for metal-on-metal bearing such explained in line 35-38. To support this explanation, the suggested reverence published should be adopted as follows: Ammarullah, M. I.; Santoso, G.; Sugiharto, S.; Supriyono, T.; Kurdi, O.; Tauviqirrahman, M.; Winarni, T. I.; Jamari, J. Tresca Stress Study of CoCrMo-on-CoCrMo Bearings Based on Body Mass Index Using 2D Computational Model. Jurnal Tribologi 2022, 33, 31–8. https://jurnaltribologi.mytribos.org/v33/JT-33-31-38.pdf
7. Rather than relying just on the predominate text as it already exists, the authors could incorporate more illustrations as figures in the materials and methods section that illustrate the workflow of the current study. It is different with flowchart for patient selection.
8. What is the baseline of patient selection? Is there any protocol, standard, or basis that has been followed? It is unclear since the patient is very heterogeneous with a small number. The resonance involved impacts the present result makes this study flaws. One major reason for rejecting this paper.
9. Important information that must be mentioned in the publication relates to the error and tolerance of the experimental equipment utilized in this investigation. As a result of the disparate findings in subsequent research by other researchers, it would be a useful discussion.
10. Outcomes must be compared to similar past research.
11. The limitation of the current study must be included at the end of the discussion section.
12. In the conclusion section, discusses future research that is required.
13. The reference is recommended to be enriched with literature from five years ago, and MDPI literature is highly recommended.
14. The authors occasionally created paragraphs in the entire document that were just one or two phrases long, which made the explanation difficult to understand. To make their explanation into a longer, more thorough paragraph, the authors should expand it. It is advised to use at least three sentences in a paragraph, with one serving as the primary sentence and the others as supporting phrases. See line 212-216 for one example.
15. After peer review, it is encouraged that a graphical abstract be included in the submission.
Author Response
Reviewer 3
Replies to Reviewer #3
We have revised the indicated parts of the manuscript according to the comments. Corrections in the newly re-revised manuscript are underlined.
Please note that your review comments are shown in italic below and our replies in non-italic.
On the comment of [As your abstract's final sentence, include a "take-home" message].
Thank you for suitable suggestion.
We described “Regular clinical surveillance is recommended for the early detection of adverse reactions to metal debris.” We have added the following sentences in before “Some cases required revisions even after the 10 years following surgery.”
On the comment of [Sort the keywords according to alphabetical order].
The keywords were sorted according to alphabetical order.
On the comment of [It is unclear whether the author's something new in this work. According to evaluation, several published studies by other researchers in the past adequately explain the issues you made in the present paper. Please be careful to highlight in the introduction section anything really innovative in this work].
Thank you for suitable suggestion.
We described “Compared with previous studies, this study reports the requirement of revision surgery for ARMD even after 10 years following THA” in discussion on line 244-245. And this study is novel as long-term outcomes of over 10 years after MOM THA in Japanese patients. We have changed the following sentences “This study aimed to investigate the long-term clinical outcomes up to minimum 10 years of total hip arthroplasty with a metal-on-metal acetabular prosthesis and describes radiological findings including magnetic resonance imaging (MRI) in Japanese patients’ with modular MOM THA” on line 49-52
On the comment of [Previous study related needs to explain in the introduction section consisting of their work, their novelty, and their limitations to show the research gaps that intend to be filled in the present study].
Thank you for suitable suggestion.
We described the research outcomes in previous study on line33-39, and added the sentences on line 39-45.
On the comment of [Line 33, minimal volumetric wear. Compared to what bearing couple? It is impossible if compared with ceramic-on-ceramic.]
We deleted the sentences line 32-34
On the comment of [The authors need supporting reference for metal-on-metal bearing such explained in line 35-38. To support this explanation, the suggested reverence published should be adopted as follows: Ammarullah, M. I.; Santoso, G.; Sugiharto, S.; Supriyono, T.; Kurdi, O.; Tauviqirrahman, M.; Winarni, T. I.; Jamari, J. Tresca Stress Study of CoCrMo-on-CoCrMo Bearings Based on Body Mass Index Using 2D Computational Model. Jurnal Tribologi 2022, 33, 31–8. https://jurnaltribologi.mytribos.org/v33/JT-33-31-38.pdf]
Thank you for suitable suggestion and advice.
We have added the reference. And we have added the sentences “Ammarullah MI et al. reported that Tresca stress value rose, and the stress distribution widened by increasing body mass index under normal walking condition using 2D finite element representing CoCrMo-on-CoCrMo hip implant” on line 42 to 45
On the comment of [What is the baseline of patient selection? Is there any protocol, standard, or basis that has been followed? It is unclear since the patient is very heterogeneous with a small number. The resonance involved impacts the present result makes this study flaws. One major reason for rejecting this paper.]
Thank you for suitable suggestion.
There were no select of patient baseline. We used this implant between 2006and 2010 in almost patients with hip disease.
On the comment of [Important information that must be mentioned in the publication relates to the error and tolerance of the experimental equipment utilized in this investigation. As a result of the disparate findings in subsequent research by other researchers, it would be a useful discussion.]
Thank you for raising important issues.
Successful positioning of the acetabular cup is a major goal in THA because of the influence of the cup position on the postoperative range of motion, impingement, wear and, finally, patient satisfaction. It is important for suitable measuring acetabular cup position after THA. And we and previous study reported the usefulness and accuracy of the three-dimensional (3D) method [ref 31.32]. In this study, no CT data was included. In the previous study that reported the comparison between the accuracy of anteroposterior radiographs (2D) and that of three-dimensional computed tomography (3D-CT) scans found a mean difference of 0.6° for inclination and a mean difference of 0.1° for anteversion between 3D CT and plain radiographic measurements. The means for absolute differences were 3.1° for inclination and 7.2° for anteversion [ref.32]. We believe that the assessment of cup position is acceptable with a 2D method similar to that of previous reports. Patient-specific factors such as contractures or leg rotation that depend on patient compliance impede the comparability of radiographic images. We calculated the ratio of the femoral offset on the operated hip to the contralateral hip for minimizing such projection errors.
We have added the sentences on line 229-240 and references [31.32].
On the comment of [Outcomes must be compared to similar past research.]
Thank you for suitable suggestion.
We described outcomes with comparison with similar past research on line 197-199, 206-213 and 226-227.
On the comment of [The limitation of the current study must be included at the end of the discussion section.]
Thank you for suitable suggestion.
We described the limitation on line 227-229.
On the comment of [In the conclusion section, discusses future research that is required.]
Thank you for suitable suggestion.
We described the conclusion section as future research on line 244-247.
On the comment of [The reference is recommended to be enriched with literature from five years ago, and MDPI literature is highly recommended.]
Thank you for suitable suggestion.
We added the references [5]. And we have added the sentences “Previous study reported that the use of cobalt chromium molybdenum (CoCrMo)-on-CoCrMo is ability to increase contact pressure by more than 47% compared to titanium alloy (Ti6Al4V) metal-on-metal couple bearings in the model based on finite element simulation” on 39-42.
On the comment of [The authors occasionally created paragraphs in the entire document that were just one or two phrases long, which made the explanation difficult to understand. To make their explanation into a longer, more thorough paragraph, the authors should expand it. It is advised to use at least three sentences in a paragraph, with one serving as the primary sentence and the others as supporting phrases. See line 212-216 for one example.]
Thank you for raising important issues.
We changed sentence on line 220-225.

Round 2
Reviewer 3 Report
Good job.